# Skin Barrier Abnormalities and Immune Dysfunction in Atopic Dermatitis

**DOI:** 10.3390/ijms21082867

**Published:** 2020-04-20

**Authors:** Gabsik Yang, Jin Kyung Seok, Han Chang Kang, Yong-Yeon Cho, Hye Suk Lee, Joo Young Lee

**Affiliations:** 1Department of Pharmacology, College of Korean Medicine, Woosuk University, Jeonbuk 55338, Korea; yangboncho@gmail.com; 2BK21plus Team, College of Pharmacy, The Catholic University of Korea, Bucheon 14662, Korea; jinkyung.seok@gmail.com (J.K.S.); hckang@catholic.ac.kr (H.C.K.); yongyeon@catholic.ac.kr (Y.-Y.C.); sianalee@catholic.ac.kr (H.S.L.)

**Keywords:** atopic dermatitis, immunity, skin, homeostasis

## Abstract

Atopic dermatitis (AD) is a common and relapsing skin disease that is characterized by skin barrier dysfunction, inflammation, and chronic pruritus. While AD was previously thought to occur primarily in children, increasing evidence suggests that AD is more common in adults than previously assumed. Accumulating evidence from experimental, genetic, and clinical studies indicates that AD expression is a precondition for the later development of other atopic diseases, such as asthma, food allergies, and allergic rhinitis. Although the exact mechanisms of the disease pathogenesis remain unclear, it is evident that both cutaneous barrier dysfunction and immune dysregulation are critical etiologies of AD pathology. This review explores recent findings on AD and the possible underlying mechanisms involved in its pathogenesis, which is characterized by dysregulation of immunological and skin barrier integrity and function, supporting the idea that AD is a systemic disease. These findings provide further insights for therapeutic developments aiming to repair the skin barrier and decrease inflammation.

## 1. Introduction

Atopic dermatitis (AD) is a common chronic relapsing inflammatory skin disease that repeatedly passes through the following stages: exacerbation and improvement. AD manifests as severe pruritus, distinct shapes and distributions of skin lesions, and genetic factors that show a personal or familial history of atopic diseases. AD is the first stage in the Allergic March, which is the sequential occurrence of allergic diseases, such as asthma and allergic rhinitis with increasing age [1]. AD typically starts in infancy or early childhood and improves or disappears with age; however, approximately 10% of patients experience symptoms up to adulthood. If continued until adolescence or adulthood, the illness is usually categorized as severe AD; such patients often have severe problems in terms of aesthetics and social adaptation, in addition to distress from the dermatitis itself. Symptoms of AD include edema, xerosis, excoriations, erythema, oozing, erosions, crusting, and lichenification, but these vary from person to person [2]. 

AD has a complex etiology including genetic, immunological, and environmental factors that cause skin barrier abnormalities and immune dysfunctions (Figure 1), which are considered crucial to the pathogenesis of AD [3]. Research is underway on the close connection between genetic mutations and AD occurrence. Major contributors to the pathogenesis of skin barrier abnormalities in AD include decreased filaggrin, ceramides, and antimicrobial peptides; increased serine protease (SP); decreased SP inhibitors; and disordered tight junctions [4]. Pathogenesis related to immune dysfunction in AD involves an increase in serum immunoglobulin E (IgE) levels, sensitization to allergens, predominance of Th2 cytokines, an increase in T cells expressing cutaneous lymphocyte-associated antigen, an increase in FcεRI expression in inflammatory dendritic epidermal cells and Langerhans cells, and increased expression of thymic stromal lymphopoietin (TSLP) (Figure 1). Recently, in both the skin barrier and immunological aspects, genetic mutations related to AD have been discovered. With respect to the skin barrier, there is typically a mutation in the filaggrin gene (*FLG*). There are also single nucleotide polymorphisms in the SP inhibitor *SPINK5* and the SP *KLK7*, and mutations in the tight junction protein claudin 1. In terms of immunological responses, there are mutations in the IgE receptor FcεRb, the innate immunity-related genes *NOD1* and *-2* and *TLR2, -4* and *-9* and mutations in the acquired immunity-related genes *IL-4, -5, -9, -10, -12, -13, -18*, and *-31* and *TSLP* [4]. 

## 2. Skin Barrier Formation and Function

The most important function of the skin is to provide an effective barrier between the internal and external environments of an organism. Thus, the skin acts as an interface between the organism and its external environment, providing both protection and support to the organism it encloses [5]. Our understanding of the skin barrier is continuously evolving, in parallel with advances in research methods [6]. The epidermal barrier serves three primary functions: limiting passive water loss, restricting environmental chemical absorption, and preventing microbial infection [7]. The epidermal barrier provides an outside-inside barrier that protects against mechanical, chemical, and microbial injury through the formation of terminally differentiated keratinocytes, a process termed keratinization [8]. During keratinization, epidermal cells progressively mature from the basal epidermal layers to form flattened cells of the stratum corneum (SC) [8]. Within the epidermis, keratinocyte proliferation is restricted to the basal cell layers. After mitosis in the basal layer, keratinocytes differentiate and migrate through the epidermis towards the SC. The differentiation process yields several keratinocyte layers within the epidermis: the stratum basale, stratum spinosum, stratum granulosum, and SC. Distinct marker genes are expressed by keratinocytes at each of the differentiation stages [9]. As the outermost layer of the skin, with a thickness of 10–20 μm, the SC is the primary mediator of the epidermal permeability barrier, accounting for over 90% of the functionality of the skin [6]. Therefore, proper development and maintenance of the SC are essential for maintaining its remarkable ability to defend the body against both chemical and microbial attacks and dehydration [10]. A major defensive function of the skin is to maintain homeostasis by preventing the uncontrolled loss of water, ions, and serum proteins. A diverse set of strategies is used by the SC to maintain epidermal integrity, including enzymatic reactions, commensal bacterial colonization, immune signaling, antimicrobial lipids and peptides, low pH, and natural moisturizing factors [8]. The complex tissue of the SC supports execution of these strategies and is composed of corneocytes and a matrix of intercellular lipids (ceramide, cholesterol, and free fatty acids), with both components derived from the terminal differentiation process of keratinocytes [11]. 

Considerable efforts have been made to elucidate the full structure, function, and biochemistry of the SC. Approximately three decades ago, Elias proposed the “brick and mortar model”, in which corneocytes (bricks) are embedded in a continuous matrix of specialized intercellular lipids (mortar) (Figure 2) [8]. The corneocytes are responsible for protection against chemical and mechanical injury, with the lipid matrix providing the essential component of the water barrier [8]. The bulk of the mechanical resistance offered by the epidermal barrier is due to corneocytes. A protein shell termed the corneocyte envelope surrounds each corneocyte; its components include loricrin, involucrin, and filaggrin. Beyond the corneocyte and in immediate contact with it sits the corneocyte lipid envelope, which is a structure of specialized lipids. These lipids and protein-rich corneocytes are critical for the formation of the functional skin barrier. Thus, the barrier function of the normal epidermis is a product of the quality of its brick and mortar components [12].

## 3. Skin Barrier Abnormalities in Atopic Dermatitis

Accumulating evidence supports a permeability barrier dysfunction in AD. Decreased levels of total ceramides and bound ceramides in the SC have been reported [13]. In addition, an abnormal expression of epidermal differentiation-related molecules, such as filaggrin, loricrin, and involucrin, has been demonstrated in AD patients [14,15], and these molecules are expected to affect permeability barrier homeostasis. 

### 3.1. Lipids

Decreases in ceramide in both lesional and non-lesional skin of patients with atopic dermatitis are uniquely observed, especially in those with filaggrin abnormalities. Moreover, it has been reported that the ratio of ceramide and cholesterol is reduced in these patients [16]. In the stratum corneum of atopic dermatitis patients, an increased pH and an elevated serine proteinase activity promote inactivation and degradation of acid sphingomyelinase and β-glucocerebrosidase, which are the necessary enzymes for ceramide synthesis [17]. An elevated serine proteinase activity reduces lamellar body secretion through plasminogen activator type 2 (PAR2) signaling and results in the abnormal transfer of various substances that are secreted from the lamellar body. This is ultimately related to the reduced stratum corneum reported in patients with atopic dermatitis [18]. In addition, cytokine cascades that are observed in various skin diseases associated with atopic dermatitis and skin barrier abnormalities reduce the synthesis of ceramide by increasing interferon alpha (IFN-α) levels [17]. In the lesions of patients with atopic dermatitis, the chain lengths of ceramide, free fatty acids, and esterified fatty acids are also shortened, which causes abnormalities in epidermal lipid organization and results in epidermal barrier permeability abnormalities [19]. In patients with chronic atopic dermatitis, increased IFN-α decreases two fatty acid elongases (i.e., elongation of very long chain fatty acids protein (ELOVL) 1 and ELOVL4), resulting in shorter N-acyl chain lengths of free fatty acids and ceramides [17]. The excessive increase in kallikrein activity may also lead to these changes in lipid structure by inducing the degradation of elongation of very long chain fatty acids protein (ELOV) [17].

### 3.2. Filaggrin

Filaggrin is an important structural protein that is responsible for the keratinization, moisturization, and antimicrobial peptide functions of the skin [20]. Filaggrin is a major component of keratohyalin granules (KGs), which are membraneless protein deposits that eventually contribute to the formation of a flattened, dead cell barrier at the skin surface [21]. The phase-separation property of filaggrin family proteins is important for KG assembly and disassembly [21]. Genetic abnormalities related to filaggrin are known to be strongly associated with atopic dermatitis [22]. In particular, filaggrin deficiency has been reported to cause atopic dermatitis at an early age, increase the sensitivity and severity of allergies, and increase infection vulnerability [23]. Filaggrin can be broken down into free amino acids and converted into urocanic acid (UCA), which maintains the acidity level in the skin, and pyrrolidine carboxylic acid (PCA), which acts as a natural moisturizer. Filaggrin abnormalities are closely related to transepidermal water loss (TEWL) and dry skin in patients with atopic dermatitis. A truncated mutation in filaggrin results in alteration in phase-separation dynamics and is linked to skin barrier disorders [21]. Thus, characterizing the genetic abnormalities in filaggrin is important for understanding the outside-inside mechanism in atopic dermatitis (Figure 3). However, 40% of filaggrin mutation carriers do not develop atopic dermatitis, and filaggrin mutations are only found in 15%–50% of patients with atopic dermatitis [22]. Since genetic abnormalities in filaggrin alone do not explain all the skin barrier dysfunctions of atopic dermatitis, further research is needed to clarify the contribution of environmental and other possible factors to this condition. Recent studies have shown that in addition to filaggrin abnormalities, abnormalities in the filaggrin-like proteins, hornerin and Filaggrin family member 2 (*FLG2*), are also involved in either the lesional or non-lesional skin barrier symptoms of atopic dermatitis [24]. Even patients with atopic dermatitis who do not harbor genetic defects related to filaggrin can have decreased filaggrin levels later, suggesting that filaggrin is closely related to the mechanism underlying the development of atopic dermatitis [25].

### 3.3. Tight Junctions (TJs)

TJs are extremely complex intracellular barriers that selectively control the cellular permeability of soluble materials. As a skin barrier, TJs exist in the cell membranes of keratinocytes of the epidermal granule layer, and thus, these structures act as a second physical barrier in the epidermis [18]. Impaired TJs are attributed to abnormal skin barrier function in AD [26]. Knockout of claudin-1 (CLDN1), which is the most important adhesion protein in TJs, in mice, results in a critical lethal epidermal barrier defect, which highlights the importance of epidermal TJs and CLDN1 [27]. Reduced CLDN1 expression in the nonlesional areas of patients with atopic dermatitis and the consequent TJ abnormalities have been reported [28,29]. Yuki and colleagues stated that abnormalities in TJs adversely affect epidermal lipids and metabolic processes associated with filaggrin [30]. 

## 4. Immune Dysfunction in Atopic Dermatitis

The pathology of AD is accompanied by an imbalance in immunity involving Th1, Th2, and Treg cells, culminating in alterations in Th1- and Th2-mediated immune responses and IgE-mediated hypersensitivity [31]. The expression levels of the Th2 cytokines IL-4 and IL-13 are elevated in AD lesions. Dupilumab, a monoclonal antibody against the IL-4/IL-13 receptor, was effective for AD treatment in clinical studies, suggesting a key role of Th2 cytokines in the pathology of AD [32]. In an AD mouse model, the Treg cell population with a Th2 cytokine profile is increased, showing that pathogenic Treg cells are increased to exacerbate AD symptoms [33]. The immune system in AD becomes more heterogeneous and complex with the activation of other immune cells, such as Th22 and Th17 cells [34].

## 5. TSLP Regulates Immune Responses in Atopic Dermatitis

TSLP belongs to the cytokine family, is closely related to IL-7, and is expressed primarily by epithelial cells in the skin, gastrointestinal tract, and lung [35]. Keratinocytes in AD skin express high levels of TSLP, contributing to the initiation and exacerbation of immune responses in the skin (Figure 4) [35]. TSLP induces the activation of dendritic cells to interact with TSLP receptor (TSLPR) and interleukin-7-receptor subunit alpha (IL-7Rα), leading to the activation of intracellular signaling pathways, such as signal transducer and activator of transcription (STAT)-5 [36]. TSLP induces the expression of OX40 ligand (OX40L) in dendritic cells, which in turn induces the differentiation of naive CD4(+) T cells into Th2 cells to generate the Th2 cytokines IL-4, -5, and -13, with downregulation of IL-10 and IFNγ [37,38,39]. TSLP/OX40L-induced Th2 responses are critical for the development of atopic dermatitis. In addition, TSLP has been shown to activate other innate immune cells, such as natural killer T cells and basophils and to modulate B cell maturation [40]. Therefore, TSLP is considered an efficient therapeutic target to treat AD (Figure 4). Downregulation of TSLP expression by dieckol, a phlorotannin compound, is well corroborated by its efficacy in improving AD-like skin symptoms in a house dust mite-induced AD model with NC/Nga mice (Figure 4) [41]. Similarly, phloxine O, a cosmetic dye, reduced TSLP expression in keratinocytes and mouse skin, correlating with alleviation of AD-like symptoms and a decrease in serum IgE and histamine levels in mice (Figure 4) [42]. These results show the positive relationship between the downregulation of TSLP expression and the treatment of AD symptoms. 

## 6. The Involvement of ILC2s in the Pathogenesis of Atopic Dermatitis

Innate lymphoid cells (ILCs) are derived from common lymphoid progenitors and belong to the lymphoid lineage. However, ILCs do not express antigen-specific receptors or myeloid or dendritic cell markers. ILCs play an important role in innate immune responses to infection and the regulation of inflammation and metabolism. ILCs include natural killer (NK) cells, noncytotoxic groups of ILCs, and lymphoid tissue-inducer (LTi) cells. ILCs are classified based on transcriptional profiles, effector cytokines, and potential effector functions (Table 1). Among ILCs, Group 2 ILCs (ILC2s) have recently drawn much attention because they participate in type 2 immunity. ILC2s respond to local Th2 antigens derived from helminth and viral infection and produce type 2 cytokines, such as interleukin (IL)-4, IL-5, IL-9, and IL-13, leading to the allergic inflammation process [43,44]. ILC2s are distributed in various tissues, including skin, lung, liver, small intestine, bone marrow, spleen, and adipose tissue, contributing to the regulation of Th2 immune responses, tissue homeostasis and repair, and metabolism. 

Dysregulation or chronic activation of ILCs leads to the exacerbation of allergic inflammatory diseases, including atopic dermatitis. Although ILC2s are present in healthy human skin, the ILC2 population is increased in the AD lesional skin of patients [45]. In transgenic mice expressing interleukin 33 with atopic dermatitis symptoms, ILC2s were greatly increased in the lesional skin, regional lymph nodes, and peripheral blood [46]. ILCs in the skin are activated by TSLP, IL-33, and IL-25, which are highly expressed in atopic dermatitis. These findings suggest a critical role of ILC2s in allergic skin diseases, such as atopic dermatitis. 

## 7. Toll-Like Receptors and Atopic Dermatitis

It has been suggested that Toll-like receptors are associated with the pathology and severity of AD immune responses. Impaired TLR2 function in the lesional skin of AD patients disrupts the normal immune response to *S. aureus*, a commensal bacterium in the skin, thereby increasing the colonization of *S. aureus* in the lesional skin area of AD patients. In addition, TLR2 levels were reduced in macrophages or peripheral blood mononuclear cells (PBMCs) that were isolated from AD patients. TLR2 ligand stimulation induces less production of Th1/Th17 cytokines such as IFNγ, IL-12, and IL-17F, and more production of the Th2 cytokine IL-5 in macrophages or PBMCs of AD patients than in macrophages and PBMCs from non-AD patients. Although TLR2 signaling in response to *S. aureus* is important for protective immune responses during the acute and initial phases of AD, continuous activation of TLR2 promotes Th1 immune responses, leading to the exacerbation of inflammation at the later stage of AD. 

Certain strains of *Candida* species reside in atopic skin, and viruses such as herpes simplex virus may exacerbate other infections and thereby worsen the inflammatory symptoms of AD. TLR 1, 2, 6, and 9 are responsible for recognizing pathogen invasion and activating host immune responses, suggesting a possible role of TLRs in the immune responses in AD lesional skin. 

Genetic polymorphisms in TLRs alter innate immune responses to infections, making AD skin vulnerable to bacterial or viral infections. Monocytes from AD patients with heterozygous R753Q polymorphism in TLR2 showed higher production of IL-6 and IL-12 in response to TLR2 than monocytes with no mutation [47]. The frequency of the R753Q in TLR2 was significantly higher in Italian AD children (16%) than in controls (6%), with a more severe phenotype [48]. However, there is a contrary report showing no correlation between the R753Q mutation in TLR2 and the severity of AD in children from Turkey [49]. The promoter polymorphism C-1237T in the TLR9 gene was associated with impaired immunity in some cases of AD [50].

## 8. The Emerging Role of Inflammasomes in Atopic Dermatitis Symptoms

House dust mite allergens and *S. aureus* activate the NOD-, LRR- and pyrin domain-containing protein 3 (NLRP3) inflammasome, suggesting a possible role of the NLRP3 inflammasome in the pathogenesis of AD. *Dermatophagoides pteronyssinus* induces caspase-1 activation and IL-1β secretion in keratinocytes. The assembly of the inflammasome complex, which consists of NLRP3, apoptosis-associated speck-like protein containing a CARD (ASC), and caspase-1, was induced by *D. pteronyssinus*. Knockdown of NLRP3, ASC, or caspase-1 results in suppression of the secretion of IL-1β and IL-18 from keratinocytes stimulated with *D. pteronyssinus* [51]. 

The expression of NLRP3 and caspase-1 is downregulated in lesional AD skin compared with healthy skin. The expression of caspase-1 is differentially regulated by Th1 and Th2 cytokines. Th1 cytokines enhance, while Th2 cytokines reduce, the expression of caspase-1. Therefore, monocytes from AD patients have impaired IL-1β secretion upon bacterial infection [52].

Since IL-1β induces TSLP expression in an NFκB-dependent manner [53], it is interesting that inflammasome activation negatively regulates dibutyl phthalate-induced TSLP expression in keratinocytes [54]. In addition, IL-33, another cytokine produced in AD skin to promote Th2 responses, is inactivated by caspase-1 [55]. These results suggest that inflammasome activation may downregulate Th2 immune responses due to the inhibition of TSLP and IL-33 expression. 

Absent in melanoma 2 (AIM2), a double-stranded DNA receptor, forms inflammasomes that produce IL-1β. AIM2 protein expression is elevated in AD skin, suggesting its role in the regulation of AD immunity [56]. 

The relationship between single-nucleotide polymorphisms in NLRP3 and atopic dermatitis susceptibility was studied in the Swedish AD population [57]. The polymorphism rs10733113 in *NLRP3* is associated with increased total IgE antibodies in male AD patients, but not in female patients [57]. The significance of polymorphism in NLRP3 and its inflammasome components needs to be further elucidated. 

## 9. Crosstalk between the Skin Barrier and Immune System in Atopic Dermatitis

The normal intact skin barrier is considered the first host defense mechanism against microbiomes and allergens that are linked with AD and often represents a part of the innate immune system. Conversely, the host immune system and cytokines are interconnected with skin barrier protein expression and function. Therefore, there exists a crosstalk between the skin barrier and immunity in AD pathology. 

Kuo et al. reported that increased TLR2 activity strengthens TJs, while the activity of TLR2 is decreased in atopic dermatitis [58]. Enhanced TLR2 activity may help restore skin barrier function through the restoration of TJ function in normal skin. TLR2 activation induces the expression of the TJ protein CLDN1 and the antimicrobial peptides, β-defensins and cathelicidin, in normal keratinocytes. However, TLR2 expression and function are altered in the lesional skin of AD patients, resulting in decreased expression of TJ proteins and antimicrobial peptides. 

These results support the hypothesis that decreased TJ function is strongly related to the abnormal skin barrier in atopic dermatitis. A decrease in CLDN1 expression in atopic dermatitis is related to increased susceptibility to herpes simplex virus-1 infection, the expression of Th2 immune markers, and the number of serum IgEs and eosinophils [59]. This indicates that CLDN1 is also associated with the immune abnormalities of patients with atopic dermatitis and their susceptibility to infection.

In response to barrier disruption, keratinocytes produce type 2 mediators, such as TSLP, IL-25, and IL-33, that activate basophils, ILC2s, and DCs [60]. In turn, ILC2s directly target TJs and reduce the barrier integrity of human keratinocytes [61]. E-cadherin, a ligand of the lectin inhibitory receptor KLRG1, has a regulatory role in ILC2 activation in atopic dermatitis. E-cadherin suppresses IL-5 and IL-13 production by ILC2s [45]. Since downregulation of E-cadherin is correlated with filaggrin insufficiency, ILC2 activity is associated with dysregulation of skin barrier integrity [45]. 

Defects in the filaggrin gene disrupt the integrity of the skin barrier and facilitate bacterial colonization and exposure to environmental factors, including allergens, leading to a skewed polarization towards the Th2 phenotype [62]. Filaggrin mutation leads to loss of skin barrier function and an increase in the ILC2 population, promoting acute skin inflammation and the development of AD. Filaggrin mutation in mice results in spontaneous skin inflammation and increases in the ILC2 population, IL-1β production, and other cytokines related to AD in skin. IL-1β and IL-1R1 signaling plays a critical role in chronic dermatitis inflammation in Filaggrin-mutant mice because anti-IL-1β antibody treatment alleviated dermatitis symptoms [63]. Among 137 patients, IL-1β levels are elevated in corneocytes from AD patients with filaggrin mutations compared with those of AD patients without these mutations. Filaggrin mutations are correlated with a reduction in natural moisturizing factors. Filaggrin-deficient mice show enhanced expression of IL-1β and IL-1RA mRNA in skin and keratinocytes. These results suggest that there is a regulatory mechanism between filaggrin and inflammasome activity in AD [64]. 

Protease-activated receptor 2 (PAR2) promotes Th2 inflammation and pruritus, and reduces the integrity of the TJ barrier by disrupting claudin-1 and occluding proteins, suggesting a role for PAR2 in TJ expression and AD pathogenesis [65,66]. 

## 10. Development of Treatment Restoring Skin Barrier Abnormalities and Immune Dysfunction

Current mainstay treatments, including emollients, steroids, calcineurin inhibitors, and immunosuppressants, have limited efficacy with potentially serious side effects. Recent studies on the pathogenesis of AD have resulted in new therapies that target specific pathways with increased efficacy and the potential for fewer systemic side effects. Impaired TJs are attributed to abnormal skin barrier function in AD and may be an important mechanism contributing to skin inflammation and unbalanced immune responses. Crisaborole and dupilumab are two new FDA-approved therapies for AD [67,68]. Treatment of crisaborole, a phosphodiesterase-4B inhibitor blocking cytokine expression, to AD patients downregulated genes involved in inflammation (*MMP12*), Th2 (*CCL22*), Th1 (*CXCL9, CXCL10*), and Th17 (*CXCL1, CXCL2*), while it upregulated tight junction claudin 8 (*CLDN8*) [69]. Dupilumab is a monoclonal antibody for IL-4 receptor α, blocking IL-4 and IL-13 pathways, while increasing the expression of filaggrin, loricrin, claudins, and ELOVL3 to reverse skin abnormalities [70]. Of the emerging therapies, the JAK-STAT inhibitors including baricitinib, upadacitinib, PF-04965842, ASN002, tofacitinib, ruxolitinib, and delgocitinib show the most promising results by restoring skin barrier function with increased filaggrin expression, as well as reducing inflammatory signaling [71]. The aryl hydrocarbon receptor modulating agent tapinarof ameliorates skin inflammation and induces barrier proteins such as filaggrin, hornerin, and involucrin [72,73]. IL-4/IL-13 antagonists, lebrikizumab and tralokinumab, showed clinical efficacy in AD patients, demonstrating the critical role of IL-13 in AD pathology [74,75]. Since IL-4 and IL-13 downregulate the expression of skin barrier proteins, such as filaggrin, loricrin, and involucrin, it is expected that the treatment of IL-13 antagonists, such as lebrikizumab and tralokinumab, would influence the recovery of skin barrier integrity [14]. Novel targeted therapeutic strategies are likely to lead to new possibilities in generating tailored treatments and enhanced clinical efficacy to combat pediatric and adult AD. Nanoparticles (NPs) have been proposed for the topical delivery of drugs used to treat skin diseases [76]. NPs may help to reduce the adverse effects of classical drugs (e.g., topical corticosteroids), as NPs show an improved safety profile since lower doses are required, as a result of site-specific delivery. NPs have also been put forward to address the problems of poor drug solubility and limited skin permeability, thereby increasing skin bioavailability [77]. Several types of NPs have been proposed for the topical delivery of sets of different drugs useful in AD (e.g., antibiotics and corticosteroids) [78]. Our study shows that the liposomal encapsulation of X-shaped oligonucleotides that act as TLR9 agonists magnifies the *in vivo* anti-AD efficacy of topical application, enabling the penetration of the skin surface [79]. 

## 11. Conclusions

Although AD is not a life-threatening disease, it severely disrupts patients’ quality of life. The future for AD patients is optimistic as many new therapies are being developed, and the pathogenesis of AD is becoming clearer. Skin barrier abnormalities and immune dysfunction are important aspects for AD pathology. They are interconnected and influence each other to initiate and aggravate AD symptoms. The underlying mechanisms and key molecules regulating skin barrier function and immune responses in AD are uncovered. The unveiling knowledge sheds a light to develop more efficient therapies with less adverse effects for AD patients. In the near future, personalized care for patients with AD is anticipated as these treatment methods include targeted therapeutics utilizing differential strategies based on clinical phenotypes and endotypes of AD, biomarkers and molecular analyses, and patient co-morbidities and complications. 

## Figures and Tables

**Figure 1 ijms-21-02867-f001:**
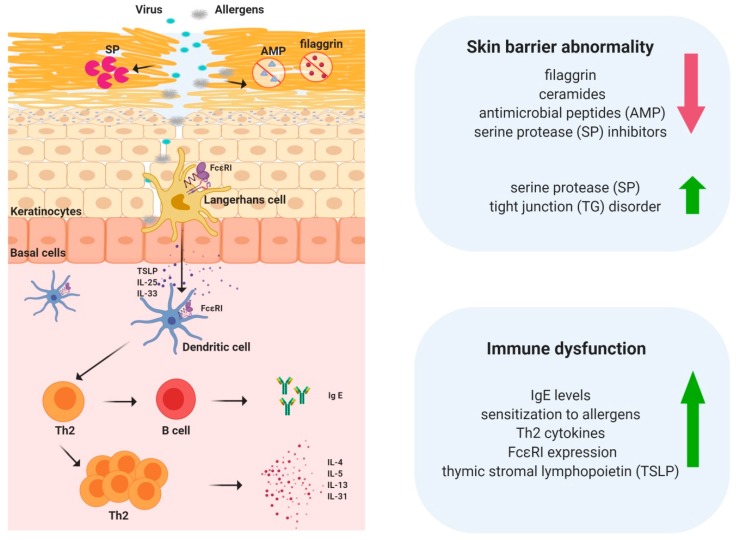
Skin barrier abnormalities and immune dysfunction are the main features of atopic dermatitis.

**Figure 2 ijms-21-02867-f002:**
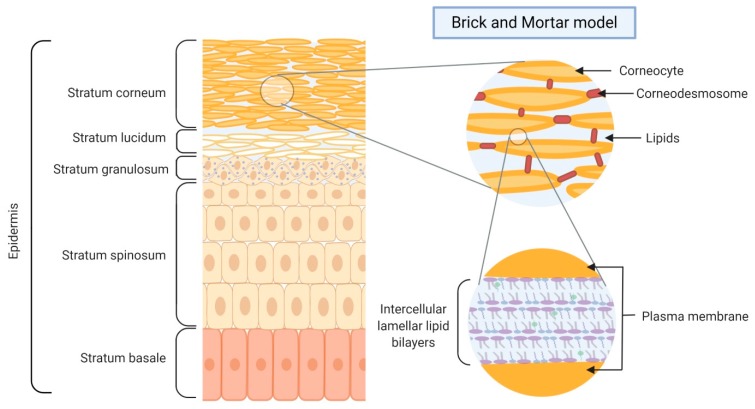
Schematic structure of the skin barrier and “brick and mortar” model.

**Figure 3 ijms-21-02867-f003:**
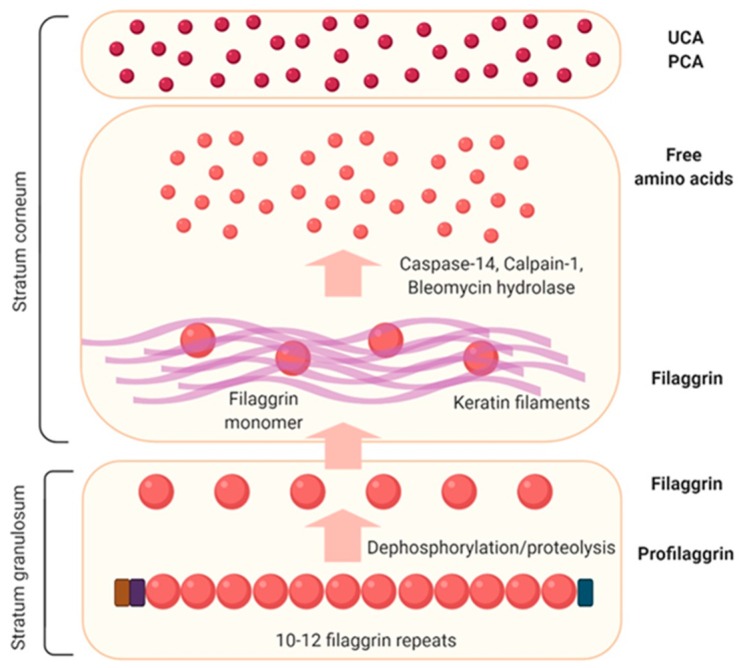
Life cycle of filaggrin. Filaggrin exists as profilaggrin within keratohyaline granules in the granular layer of the epidermis. Profilaggrin is degraded to form filaggrin during the terminal differentiation process. Then, in the upper part of the stratum corneum (SC), filaggrin is degraded into amino acids and plays a crucial role in maintaining SC hydration and pH by forming natural moisturizing factors, including pyrrolidine carboxylic acid (PCA) and urocanic acid (UCA).

**Figure 4 ijms-21-02867-f004:**
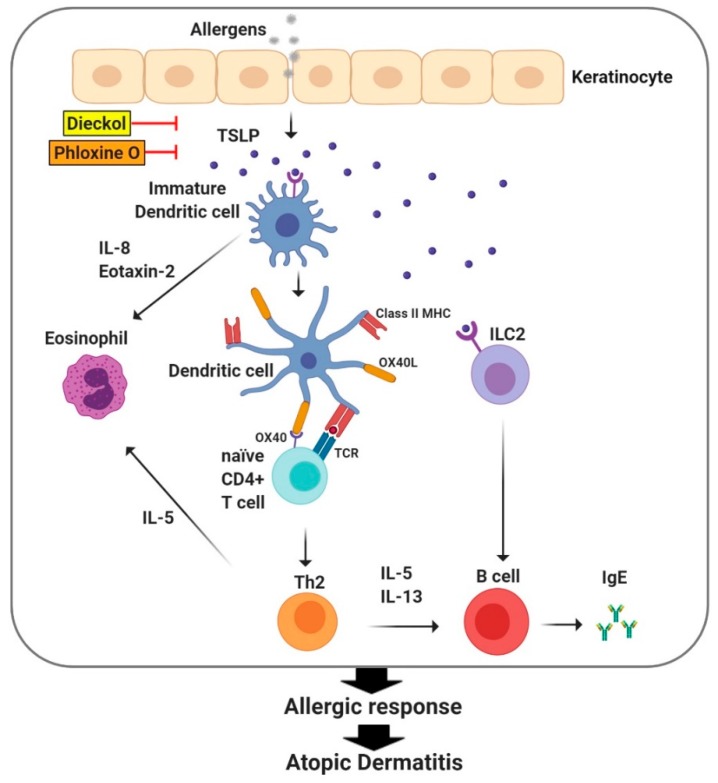
Thymic stromal lymphopoietin (TSLP) initiates the innate and adaptive phases of allergic immune responses in the skin. TSLP induces the maturation of dendritic cells to express OX40L, which in turn differentiates naive CD4+ T cells into Th2 cells to produce Th2 cytokines such as IL-4, IL-5, and IL-13, leading to the secretion of IgE from B cells. Together with the activation of Group 2 innate lymphoid cells (ILC2s), TSLP initiates the innate and adaptive immune responses of atopic dermatitis. Dieckol and phloxine O reduce atopic dermatitis-like inflammatory symptoms by inhibiting TSLP production.

**Table 1 ijms-21-02867-t001:** Classification of ILC subsets.

Cell Type		Cytokines Required for Development	Transcription Factors	Stimulating Cytokines	Cytokine Production	Biological Function
**Group 1 ILCs**	NK cell	IL-15	T-bet EOMES	IL-12 IL-18	IFN-γ TNF	Immunity to virus and cancer Chronic inflammation
	ILC1	IL-7 IL-15	T-bet	IL-12 IL-18	IFN-γ TNF	Immunity to intracellular bacteria and protozoa Chronic inflammation
**Group 2 ILCs**	ILC2	IL-7	BCL11B GFI1 EST1 GATA3	IL-25 IL-33 TSLP	IL-4 (in humans) IL-5 IL-13 AREG	Immunity to helminths Asthma and allergic disease Metabolic homeostasis
**Group 3 ILCs**	LTi cell	IL-7	RORγt	IL-23 IL-1β	IL-17A/IL-17F IL-22	Lymphoid tissue developments Intestinal homeostasis Immunity to extracellular bacteria Chronic inflammation
	NCR^-^ ILC3	IL-7	AHR RORγt	IL-23 IL-1β	IL-17A/IL-17F IL-22	
	NCR^+^ ILC3	IL-7	AHR RORγt T-bet	IL-23 IL-1β	IL-22	

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
