# Peer review of "Skin Barrier Abnormalities and Immune Dysfunction in Atopic Dermatitis"

_ijms, 2020, doi:10.3390/ijms21082867_

Round 1

Reviewer 1 Report

This review is well-written and provides a fairly good overview of what is currently known about the role of the barrier in AD. The introductory section on "skin barrier formation..." is fairly "text book" but is probably fine to leave in for those readers unfamiliar with skin structure.

I think the readers would have been interested in reading a little about current treatments for AD and how these treatments alter or reverse any of the abnormalities discussed in the review. That information would make a stronger review.

Although there are some recent citations, many of the references are from studies over 10 or 15 years old. More current references would have been helpful.

Author Response

Authors’ Response to the Reviewers comments

Manuscript number: ijms-773997

Manuscript title: Skin barrier abnormalities and immune dysfunction in atopic dermatitis

We would like to express our sincere appreciation to the reviewers’ comments to improve our manuscript. We have tried our best to resolve each of the comments and hope the revision is satisfactory. We provide our point-to-point responses here and the changes made are shown in red-colored font in the revised manuscript.

Response to Reviewer #1

Comment 1: I think the readers would have been interested in reading a little about current treatments for AD and how these treatments alter or reverse any of the abnormalities discussed in the review. That information would make a stronger review.

Response 1: We added the current treatment for AD as follows:

             Development of treatment restoring skin barrier abnormalities and immune dysfunction

Current mainstay treatments, including emollients, steroids, calcineurin inhibitors, and immunosuppressants, have limited efficacy with potentially serious side effects. Recent studies on the pathogenesis of AD have resulted in new therapies that target specific pathways with increased efficacy and the potential for fewer systemic side effects. Impaired TJs are attributed to abnormal skin barrier function in AD and may be an important mechanism contributing to skin inflammation and unbalanced immune responses. Crisaborole and dupilumab are two new FDA-approved therapies for AD [67, 68]. Treatment of crisaborole, a phosphodiesterase-4B inhibitor blocking cytokine expression, to AD patients downregulated genes involved in inflammation (MMP12), Th2 (CCL22), Th1 (CXCL9, CXCL10), Th17 (CXCL1, CXCL2) while it upregulated tight junction claudin 8 (CLDN8) [69]. Dupilumab is a monoclonal antibody for IL-4 receptor a, blocking IL-4 and IL-13 pathways while increasing expression of filaggrin, loricrin, claudins, and ELOVL3 to reverse skin abnormalities [70]. Of the emerging therapies, the JAK-STAT inhibitors including baricitinib, upadacitinib, PF-04965842, ASN002, tofacitinib, ruxolitinib, and delgocitinib show the most promising results by restoring skin barrier function with increased filaggrin expression as well as reducing inflammatory signaling [71]. The aryl hydrocarbon receptor modulating agent tapinarof ameliorates skin inflammation and induces barrier proteins such as filaggrin, hornerin, and involucrin [72, 73]. The IL-4/IL-13 antagonists, lebrikizumab and tralokinumab showed clinical efficacy to AD patients demonstrating critical role of IL-13 in AD pathology [74, 75]. Since IL-4 and IL-13 downregulate the expression of skin barrier proteins such as filaggrin, loricrin, and involucrin, it is expected that the treatment of IL-13 antagonists such as lebrikizumab and tralokinumab would influence to recover skin barrier integrity [76]. Novel targeted therapeutic strategies are likely to lead to new possibilities in generating tailored treatments and enhanced clinical efficacy to combat pediatric and adult AD. Nanoparticles (NPs) have been proposed for the topical delivery of drugs used to treat skin diseases [77]. NPs may help to reduce the adverse effects of classical drugs (e.g., topical corticosteroids) as NPs show an improved safety profile since lower doses are required as a result of site-specific delivery. NPs have also been put forward to address the problems of poor drug solubility and limited skin permeability, thereby increasing skin bioavailability [78]. Several types of NPs have been proposed for the topical delivery of sets of different drugs useful in AD (e.g., antibiotics and corticosteroids) [79]. Our study show that liposomal encapsulation of X-shaped oligonucleotides that act as TLR9 agonists, magnifies in vivo anti-AD efficacy of topical application, enabling the penetration of skin surface [80].

Comment 2: Although there are some recent citations, many of the references are from studies over 10 or 15 years old. More current references would have been helpful.

Response 2: We added recent publications while adding the current treatments for AD in response to the comment 1.

Reviewer 2 Report

This review describes some features and possible mechanisms involved in atopic dermatitis. I think that this paper can be published, however I suggest same modifications.

I think that the authors should insert a conclusion at this review. 

In line 242 the title "The emerging role of inflammasomes in atopic dermatitis symptoms" should be reported in the line 243.

Please the authors should be define some words: ELOV, FLG2, NLRP3, ASC   

In line 222 missing the point after the word "responses".

Author Response

Authors’ Response to the Reviewers comments

Manuscript number: ijms-773997

Manuscript title: Skin barrier abnormalities and immune dysfunction in atopic dermatitis

We would like to express our sincere appreciation to the reviewers’ comments to improve our manuscript. We have tried our best to resolve each of the comments and hope the revision is satisfactory. We provide our point-to-point responses here and the changes made are shown in red-colored font in the revised manuscript.

Response to Reviewer #2

Comment 1: I think that the authors should insert a conclusion at this review. 

Response 1: We added a conclusion as follows:

Conclusion

Although AD is not a life-threatening disease, it severely disrupts patients’ quality of life. The future for AD patients is optimistic as many new therapies are being developed, and the pathogenesis of AD is becoming clearer. Skin barrier abnormalities and immune dysfunction are important aspects for AD pathology. They are interconnected and influenced each other to initiate and aggravate AD symptoms. The underlying mechanisms and key molecules regulating skin barrier function and immune responses in AD are uncovered. The unveiling knowledge shed a light to develop more efficient therapy with less adverse effects for AD patients. In near future, personalized care for patients with AD is anticipated as these treatment methods include targeted therapeutics utilizing differential strategies based on clinical phenotypes and endotypes of AD, biomarkers and molecular analyses, and patient co-morbidities and complications.

Comment 2: In line 242 the title "The emerging role of inflammasomes in atopic dermatitis symptoms" should be reported in the line 243.

Response 2: In line 242 the title "The emerging role of inflammasomes in atopic dermatitis symptoms" is moved to the line 243.

Comment 3: Please the authors should be define some words: 

ELOV, FLG2, NLRP3, ASC   

Response 3: We defined the following words:

ELOV: elongation of very long-chain fatty acids protein

FLG2: Filaggrin family member 2

NLRP3: NOD-, LRR- and pyrin domain-containing protein 3

ASC: apoptosis-associated speck-like protein containing a CARD

Comment 4: In line 222 missing the point after the word "responses".

Response 4: The point was added after the word "responses" in line 222.